# Genome-Wide Identification, Characterization, Evolutionary Analysis, and Expression Pattern of the GPAT Gene Family in Barley and Functional Analysis of *HvGPAT18* under Abiotic Stress

**DOI:** 10.3390/ijms25116101

**Published:** 2024-06-01

**Authors:** Chenglan Yang, Jianzhi Ma, Cunying Qi, Yinhua Ma, Huiyan Xiong, Ruijun Duan

**Affiliations:** 1College of Eco-Environmental Engineering, Qinghai University, Xining 810016, China; lan961001@163.com (C.Y.); ys210713000109@qhu.edu.cn (J.M.); 13897238953@163.com (C.Q.); mayinhua2021@163.com (Y.M.); 2College of Agriculture and Animal Husbandry, Qinghai University, Xining 810016, China; 1995990015@qhu.edu.cn

**Keywords:** *Hordeum vulgare* L., glycerol-3-phosphate acyltransferase (GPAT), gene family, classification, evolution, expression pattern, functional analysis, abiotic stress

## Abstract

Glycerol-3-phosphoacyltransferase (GPAT) is an important rate-limiting enzyme in the biosynthesis of triacylglycerol (TAG), which is of great significance for plant growth, development, and response to abiotic stress. Although the characteristics of GPAT have been studied in many model plants, little is known about its expression profile and function in barley, especially under abiotic stress. In this study, 22 *GPAT* genes were identified in the barley genome and divided into three groups (I, II, III), with the latter Group III subdivided further into three subgroups based on the phylogenetic analysis. The analyses of conserved motifs, gene structures, and the three-dimensional structure of HvGPAT proteins also support this classification. Through evolutionary analysis, we determined that HvGPATs in Group I were the earliest to diverge during 268.65 MYA, and the differentiation of other HvGPATs emerged during 86.83–169.84 MYA. The tissue expression profile showed that 22 *HvGPAT* genes were almost not expressed in INF1 (inflorescence 1). Many functional elements related to stress responses and hormones in cis-element analysis, as well as qRT-PCR results, confirm that these *HvGPAT* genes were involved in abiotic stress responses. The expression level of *HvGPAT18* was significantly increased under abiotic stress and its subcellular localization indicated its function in the endoplasmic reticulum. Various physiological traits under abiotic stress were evaluated using transgenic Arabidopsis to gain further insight into the role of *HvGPAT18*, and it was found that transgenic seedlings have stronger resistance under abiotic stress than to the wild-type (WT) plants. Overall, our results provide new insights into the evolution and function of the barley GPAT gene family and enable us to explore the molecular mechanism of functional diversity behind the evolutionary history of these genes.

## 1. Introduction

Lipids from plants are composed of various fatty acids and their derivatives, such as lipid polyesters, glycerol-lipids, and sterols. As major components of cellular membranes, storage, extracellular protective layers, and signaling molecules, they participate in a wide range of metabolic reactions and play vital physiological roles in plant growth and development [1,2,3]. Glycerol lipids are formed by taking glycerol as the molecular skeleton and acylating at the sn-1, sn-2, or sn-3 hydroxyl groups of glycerol molecules. Their types include phospholipids, glycolipids, triglycerides, and extracellular lipids, such as cuticle and suberin [4,5,6,7]. Several intermediates, such as lysophosphatidic acid (LPA) and phosphotic acid (PA), are substrates for the generation of important glycerol lipids, such as extracellular lipid polyesters, storage lipids, and membrane lipids. In the de novo synthesis of glycerol lipids, glycerol-3-phosphate (G3P) carries out the lipoacylation under the catalysis of glycerol-3-phosphate acyltransferase (GPAT) to form LPA, and then dephosphorylates LPA to form PA under the catalysis of LPA acyltransferase (LPAAT). GPAT is an important rate-limiting enzyme in the process of glycerol lipid synthesis. It can be divided into sn-1 and sn-2 because it catalyzes the sn-1 and sn-2 positions of G3P [1,3,4,8,9]. At the same time, the metabolic fate of LPA is partly controlled by the subcellular localization of the GPAT-catalyzed reaction. Therefore, it can also be divided into three types of GPATs due to the fact that the reaction occurs in three different plant subcellular compartments, namely plastid, endoplasmic reticulum (ER), and mitochondria. Specifically, plastid GPAT is soluble, and the acyl group is transferred to the sn-1 position of G3P through acyl ACP. It belongs to the sn-1 type and is crucial for the synthesis of phosphatidylglycerol (PG) in plastid; ER and mitochondrial GPATs are membrane-bound enzymes. In addition to using acyl-ACP as an acyl donor to catalyze the sn-1 position of G3P, these two other GPATs also have additional phosphatase activity to form sn-2 MAG (sn-2 monoacylglycerol) using acyl-CoA as an acyl donor. These GPATs are classified as sn-2 type, which are mainly involved in the synthesis of extracellular lipids, such as cuticle and suberin [1,2,3,6,8,9].

The GPAT gene family has been identified in many model plants, including *Arabidopsis thaliana* [8,10], *Brassica napus* [11], *Oryza sativa* [10], *Zea mays* [12], and other plants, such as *Paeonia* [13], *Gossypium* [14], and *Perilla frutescens* [15]. It has been confirmed that the members of the GPAT gene family have been widely described for their role in male fertility [16,17,18,19,20,21], seed development [22,23,24,25,26,27,28,29,30], biotic and abiotic stress tolerance [31,32,33,34,35,36,37,38,39,40,41,42,43], and cuticle or suberin production [5,22,44,45,46,47,48,49]. At present, the research on the GPAT gene family is the best in the model plant Arabidopsis. In Arabidopsis, ten members of the GPAT gene family, *AtS1* and *AtGPAT1*–*9*, have been identified [8,10]. According to evolutionary distance, they are divided into three subgroups, namely *ATS1*, *AtGPAT9*, and *AtGPAT1*–*8* [3]. AtS1 encodes an enzyme in the form of soluble chloroplast and belongs to sn-1 type GPTA. It can control cold tolerance by controlling the fatty acid composition at the sn-1 position of PG in the plastid membrane, thereby affecting the membrane fluidity of plant aerial tissues [40]. The other nine genes (*AtGPAT1* to *AtGPAT9*) encode mitochondrial membrane-related enzymes (AtGPAT1–AtGPAT3) or ER-related enzymes (AtGPAT4-AtGPAT9) and have sn-2 acyltransferase activity. AtGPAT9 plays a role in plant membrane, triglyceride biosynthesis, and intracellular glycerol synthesis [50,51,52]. AtGPAT1 is involved in tapetal differentiation and male fertility, and its destruction leads to the stagnation of pollen development, revealing its role in pollen development [16]. AtGPAT4-8 is involved in the synthesis of extracellular lipid barrier polyesters [8,17], and AtGPAT5 contributes to the synthesis of suberin in seed coat [5,22,44].

The function of *GPAT* genes under abiotic stresses such as drought, salinity, low temperature, and high temperature has been studied in many plants [33,34,36,37,38,40,41,43,53]. It is reported that the overexpression of the *AtS1* gene improves the tolerance of Arabidopsis seedlings to low temperature stress, while the transformation of the squash GPAT gene into tobacco leads to a reduction in low temperature tolerance [31,40]. In rice, overexpression of the *AtS1* gene increased the unsaturation of fatty acids in phosphatidylglycerol (PG) and improved photosynthetic rate and growth at low temperatures [32]. Overexpression of the *GPAT* gene also enhances chilling tolerance in tomato plants [33]. However, antisense-mediated deletion of the chloroplast *LeGPAT* gene alleviated heat stress injury in transgenic tomato plants [33,35]. Overexpression of the *AmGPAT* gene in *Ammopiptanthus mongolicus* can improve the tolerance of transgenic Arabidopsis to cold and other oxidative stress by increasing the level of unsaturated fatty acids in phosphatidylglycerol [40]. In another study, the *LpGPAT* gene in *Lilium pensylvanicum* was proved to be related to low-temperature stress [36]. Overexpression of the *SsGPAT* gene in *Suaeda salsa* enhanced salt tolerance in Arabidopsis [37],while loss-of-function of *ATS1* enhances Arabidopsis salt tolerance [43]. *SpGPAT5* in *Sarracenia purpurea* can functionally replace *AtGPAT5* and contributes to the plant’s tolerance of high humidity [41]. In conclusion, GPAT plays an important role in the process of plant adaptation to abiotic stress.

Although the GPAT family has been studied in many model plants, little is known about its expression profile and function in barley, especially under abiotic stress. Based on barley genome data, this study identified members of the barley GPAT (HvGPAT) family, analyzed its characterization (i.e., its gene structure, subcellular localization, protein conserved domains, phylogenetic relationships, and evolution), explored the expression specificity of *HvGPATs* in various tissues and under abiotic stress conditions, and studied the function of the *HvGPAT18* gene through transgenic Arabidopsis, so as to lay the foundation for the analysis of GPAT function and stress resistance regulation and also provide a theoretical reference for abiotic stress resistance gene mining and abiotic stress resistance breeding in barley.

## 2. Results

### 2.1. Genome-Wide Identification and Analysis of HvGPAT Genes 

Using GPAT protein sequences from Arabidopsis and rice as query sequences, the barley candidate genes were obtained using BLASTP and HMM programs. After sequence merging and removing redundancy, domain validation was performed using Pfam, NCBI-CDD, and Smart databases. Subsequently, a total of 22 barley GPAT genes were identified and named HvGPAT1 to HvGPAT 22 (Table 1 and Appendix A), based on the sequence location of these genes on barley chromosomes. The identified GPAT proteins range in size from 154 (HvGPAT9) to 556 (HvGPAT17) aa (amino acids). The predicted molecular weight of HvGPAT proteins ranges from 17,324 to 61,130.50 Da, and the theoretical PI ranges from 6.61 to 10.3. Except for HvGPAT8, HvGPAT16, HvGPAT17, and HvGPAT20, most HvGPAT proteins have high instability (instability value > 40). In addition, transmembrane domain prediction showed that, except for six proteins (HvGPAT3/HvGPAT4/HvGPAT9/HvGPAT10/HvGPAT15/HvGPAT17), most HvGPAT proteins have one or more hypothetical transmembrane domains (TMD), indicating that they are membrane proteins. Protein subcellular localization prediction showed that 22 HvGPAT proteins were predicted to be located in mitochondria, ER, and chloroplast (Table 1 and Appendix A). 

### 2.2. Chromosome Localization Analysis of HvGPAT Genes

According to the annotation information of the barley genome, 21 *HvGPAT* genes were unevenly and loosely distributed on seven barley chromosomes; however, the chromosome position of the *HvGPAT22* gene is unknown (Figure 1). There were six *HvGPAT* genes on chr1 and chr3 chromosomes, and only 1–3 *HvGPAT* genes were distributed on the other chromosomes, indicating that HvGPAT gene-distributed events may mainly occur on barley chromosomes 1 and 3 during evolution.

### 2.3. Phylogenetic and Evolutionary Analysis of the HvGPAT Gene Family

To determine the phylogenetic relationships of HvGPAT gene family members, 50 GPAT protein sequences from three species (*Arabidopsis thaliana*, *Oryza sativa*, and *Hordeum vulgare*) were selected and compared using mega 7.0 software to create a phylogenetic tree (Figure 2A). Phylogenetic analysis revealed that a total of 50 GPAT genes can be divided into three groups (I, II, III), with the latter Group III subdivided further into three subgroups (IIIa, IIIb, IIIc) (Figure 2A and Appendix A). Group I contains six HvGPATs, three OsGPATs, and an AtS1. Group II contains five HvGPATs, two OsGPATs, and an AtGPAT9. Group III can be further classified into three subclades. Subclade III-a comprises five HvGPATs, nine OsGPATs, and three AtGPATs (AtGPAT1–3); subclade III-b contains three HvGPATs, one OsGPAT, and two AtGPATs (AtGPAT5 and 7); and subclade III-c includes three HvGPATs, three OsGPAT, and three AtGPATs (AtGPAT4, 6 and 8). 

To understand the evolutionary history of the barley gene family, we constructed an evolutionary history tree (Figure 2B). The results showed that a distant evolutionary relationship exists among different clades of GPAT genes in barley. Most of the species studied present only one GPAT9 and one soluble GPAT, except for *Brassica rapa*, *Zea mays*, and *Hordeum vulgare*, which means that gene duplication events may have occurred in these exceptional species. In Group I (soluble GPAT), HvGPAT3 first differentiated from TaGPAT9 and TaGPAT16 at 268.25 MYA, and was followed by BdGPAT16 (*Brassica napus*), OsGPAT15 (*Oryza sativa*), and ZmGPAT2 (*Zea may*). HvGPAT15 in Group I evolved later than other genes. In Group II (GPAT9), HvGPAT4 and HvGPAT11 differentiated before 169.84 MYA, and TaGPAT3, TaGPAT11, and TaGPAT18 differentiated at the same time. HvGPAT21 finally differentiated at 180.65 MYA. Group III (GPAT1–8) is subdivided into three subcategories: IIIc (GPAT4, GPAT8, and GPAT6), IIIb (GPAT5 and GPAT7), and IIIa (which includes GPATs 1, 2, and 3). In IIIa, GPAT1, GPAT2, and GPAT3 are closely related and may originate from replication events in vascular plants. In this subclass, we observed that monocotyledons and dicotyledons have no obvious relationship in evolutionary time. The GPAT genes of *Triticum aestivum*, *Brassica napus*, and *Oryza sativa* evolved first; GPAT5 and GPAT7 in subclass IIIb may have also been caused by independent and lineage-specific replication events. This subclass contains a set of sequences, only including *Zea may*, *Brassica napus,* and *Triticum aestivum*, without the representative GPAT sequence of *Hordeum vulgare*. The three GPAT (HvGPAT6, HvGPAT14, and HvGPAT19) genes from barley are more phylogenetically related to AtGPAT4, AtGPAT8, and AtGPAT6 (IIIc subclass), with the latter subclass differentiated at 86.83MYA, indicating that IIIc subclass is the youngest form of GPAT in terrestrial plants. 

### 2.4. Intron–Exon Patterns and Conserved Motifs Analysis of HvGPAT Genes

GPAT enzymes have four highly conserved domains that are thought to play a role in acyltransferase activity [54]. Multiple sequence alignment analysis (Figure 3A) showed that four motifs of HvGPAT proteins were highly conserved, with acyltransferase motifs I and III being the most conserved regions in HvGPAT family enzymes. Almost all enzymes in group II and group III have the characteristics of HXXD and PEGT-X. This result is consistent with the previous research results, indicating that motif II may be involved in the binding of acyl receptors LPA and G3P. The mutation of proline to leucine has little effect on GPAT enzyme activity, but replacing proline with glutamate or lysine can eliminate GPAT activity. It can be seen from Figure 3A that the conserved proline sites of HvGPAT3 and HvGPAT15 in group I are replaced with alanine and histidine, respectively, which is likely to affect their activity. 

The exon/intron pattern in the gene family could be significant during the evolutionary process and may be associated with gene function [11]. Therefore, we examined the gene structures of *HvGPAT* members using the GSDS server by comparing the full-length *HvGPATs* CDS and their genomic DNA sequences (Figure 3B). The results showed that the gene structure of the *HvGPATs* could be divided into two obvious categories, in which the 14 genes of group III contain few introns, mostly between one and two introns. On the contrary, the other genes of group I and group II have more introns, ranging from 4 to 15. Among them, *HvGPAT1*, *HvGPAT8*, and *HvGPAT20* have the most introns, consisting of 15, 8, and 13 introns, respectively. The gene structure of the same group member is similar. In addition, the intron phage of all *HvGPATs* was also studied (Figure 3B). The results indicated that the intron phase pattern is very variable in different *HvGPATs*. Phase 0 was predominant across genes with only one intron, which is the case of most *HvGPATs* in group III. Intron phase 0, 0, 0, 1, 0, 0 is significantly conserved in the hypothetical *GPAT9* gene, and these genes are included in group II along with Arabidopsis *AtGPAT9*. For soluble GPAT (AtS1), the intron phase pattern is not fixed.

Using the online MEME program to analyze the protein sequences of HvGPAT members, we found that the domain composition of HvGPATs has many forms. As can be seen from Figure 3, groups I, II, and III have different motif components. For example, group I is special and very different from the other two groups. In this group, HvGPAT9 and HvGPAT15 proteins do not have any of the 10 conserved motifs we searched. The conserved motifs of other GPAT proteins in group I in are motif-1, motif-6, motif-8, and motif-9. The conserved motifs of HvGPAT2 and HvGPAT21 were not found, and the main conserved domain of other GPAT proteins in group II was motif-1. Group Ⅲ is the most complex group of conserved motifs, and is mainly composed of motif-1, motif-2, motif-3, motif-4, and motif-5. We also used a three-dimensional structural simulation method to obtain the spatial structure of the barley GPAT gene family at the full-length amino acid level (Appendix A) and domain level (Appendix A), respectively, which also supports classification. In general, the motif composition, gene structure, and protein structure among HvGPAT members in the same group are similar, which can further prove the reliability of classification and evolutionary analysis.

### 2.5. Expression Profiles of HvGPAT Genes in Different Tissues

Based on FPKM (Fragments Per Kilobase Million) values from the IPK database, the expression heat map of *HvGPAT* genes in various tissues was established (Figure 4). The results showed that these 22 *HvGPAT* genes have a typical feature, that is, they are almost not expressed in INF1. At the same time, from the phylogenetic relationship, it can be concluded that the *HvGPAT* family has two different expression patterns. In the A group with relatively high expression levels, except for *HvGPAT18,* which is not expressed in INF1, the expression level of *HvGPAT18* in each tissue is generally higher than that of other genes, while *HvGPAT14* is specifically highly expressed in EMB, CAR5, and LOD, *HvGPAT21* is relatively high in ROO2, and *HvGPAT3* is high in SEN. In the low-expression B group, except for the relatively high expression level of *HvGPAT15* in IFN2, the expression levels of other genes in various tissues are very low.

### 2.6. The Promoter Analysis of HvGPAT Genes

The cis-acting elements of *HvGPAT* gene promoter sequences were analyzed by Plantcare software (http://bioinformatics.psb.ugent.be/webtools/plantcare/html/ accessed on 18 September 2021) (Figure 5). A total of 14 cis-acting elements related to stress response and plant hormones were identified. Among them, there are five elements related to stress response, including anaerobic inducible element (ARE), cis-acting element involved in defense and stress response (TC-rich repeat), low-temperature inducible element (LTR), and drought inducible response element (MBS). *HvGPAT* genes containing LTR and MBS elements account for 27.3%. There are 10 elements related to plant hormones, including abscisic acid response element (ABRE), auxin response element (TGA-element, AuxRR-core), methyl jasmonate response element (TGACG motif, CGTCA motif), ethylene response element (ERE), gibberellin response element (TATC-box, P-box, GARE-motif), and salicylic acid response element (TCA-element), while *HvGPAT* genes containing abscisic acid response elements accounted for 86.4%.

### 2.7. Expression Profiles of HvGPAT Genes under Abiotic Stress 

Barley seedlings were treated with drought, low temperature, and salt stress for two days; after two days of normal culture, the expression levels of 11 genes (*HvGPAT1, HvGPAT3*, *HvGPAT4*, *HvGPAT5*, *HvGPAT6*, *HvGPAT8*, *HvGPAT12*, *HvGPAT14*, *HvGPAT17, HvGPAT18*, and *HvGPAT20*) in roots and leaves were analyzed (Figure 6). The results showed that, under low temperature, drought, and salt stress, the expression levels of *HvGPAT3*, *HvGPAT5*, *HvGPAT6,* and *HvGPAT12* genes significantly increased in leaves and roots, while the expression levels of *HvGPAT5*, *HvGPAT14*, *HvGPAT17,* and *HvGPAT18* genes increased in leaves. The expression level of *HvGPAT4* increased in roots after cold treatment and also increased in roots and leaves after salt stress. The expression level of *HvGPAT8* was up-regulated in leaves after cold stress, and up-regulated in roots and leaves after drought stress. The expression levels of *HvGPAT14* and *HvGPAT20* were up-regulated in roots after cold stress. The expressions levels of *HvGPAT17* and *HvGPAT18* were up-regulated in roots after drought stress. This shows that these *HvGPAT* genes have a strong response to low temperature, drought, and salt stress (Figure 6).

### 2.8. Overexpression of HvGPAT18 in Arabidopsis and Its Response to Abiotic Stress

Except for INF1, the expression level of *HvGPAT18* is generally high, especially in EMB, NOD, LOD, and ROO2 (Figure 4). The expression level of *HvGPAT18* was significantly increased under abiotic stress (Figure 6) and its subcellular localization indicated their function at endoplasmic reticulum (Table 1 and Appendix A). To further explore the role of *HvGPAT18* in abiotic stress, transgenic Arabidopsis plants overexpressing this *HvGPAT18* gene were generated. Seeds from 35S::*HvGPAT18* transgenic lines (L1, L2, L3) and WT were sown on an MS basal medium and subjected to drought, salt, and cold stress treatments using mannitol, NaCl, and 4 °C, respectively. Under normal conditions, there was no difference in root length between the 35S::*HvGPAT18* transgenic line and WT. However, the transgenic root length was longer than that of WT on the MS medium with NaCl, mannitol, 4 °C (Figure 7A,C). The germination rates in the transgenic lines and WT showed no significant difference under normal cultivation conditions, while it was significantly decreased in WT under drought, cold, and salt stress, especially under drought stress (Figure 7B). Proline (PRO) is considered to be a reliable physiological indicator of stress resistance in plants, and the proline content of the transgenic line was higher than that of WT under drought and salt stress (Figure 7D). In addition, the hydrogen peroxidase (CAT) content of the three overexpressed lines was higher than that of the WT, when exposed to cold, drought, salt stress, and control (Figure 7E). These results indicate that the overexpression of *HvGPAT18* in Arabidopsis resulted in more tolerance than in WT plants under abiotic stress.

## 3. Discussion

GPAT is a gene family with a PLSC acyltransferase domain, which is related to plant growth, development, and resistance to abiotic stresses. Almost all eukaryotes have GPAT genes, and the number of GPAT members varies among different plant species. For example, 10 GPAT genes were identified in *Arabidopsis thaliana*, 18 in *Oryza sativa*, 20 in *Zea mays*, and 32 in *Brassica napus* [8,10,11,12,19]. In this study, 22 GPAT members were identified in barley and named HvGPAT1–HvGPAT22, respectively, and the number of this gene family is similar to that of monocotyledons, for example, rice and maize. In terms of physical and chemical properties, 10 Arabidopsis GPAT genes encode 376~585 amino acids, the isoelectric point ranges from 6.52~10.02, 18 rice GPAT genes encode 264~570 amino acids, the isoelectric point ranges from 6.74~10.80, 20 maize GPAT genes encode 136~595 amino acids, the isoelectric point ranges from 6.18~9.80, and 32 *Brassica napus* GPAT genes encode 203~572 amino acids, the isoelectric point range is 5.94~10.08 [10,11,12]. Comparing the encoded protein lengths and theoretical isoelectric points (154–556 amino acids, 6.61–10.34) of 22 HvGPAT proteins (Appendix A), it was found that they are highly unstable membrane proteins with hydrophilicity and hydrophobicity, similar to those of other species.

### 3.1. Classification of HvGPAT Genes

Although many studies have revealed the key role of GPATs in glycerol lipid biosynthesis, the understanding of GPATs is still limited. To further understand the functions of GPAT in glycerol lipid biosynthesis and different physiological processes, it is very important to understand their evolutionary history and diversity. In this study, we provide the general situation of barley GPAT, including their gene family members, evolutionary history, tissue expression, and abiotic stress expression analysis. Three main branches were identified in the phylogenetic tree and named Group I, Group II, and Group III. Group III is the most diverse branch, which is further subdivided into three subclasses (IIIa, IIIb, and IIIc) (Appendix A).

Soluble plastid GPAT is the first GPAT found in plants [2]. This enzyme is essential for the synthesis of chloroplast glycerides, which are mainly converted into galactolipids, which is the main structural and functional component of photosynthetic membranes [55]. The analysis of AtS1 gene expression in barley showed that soluble GPAT was mainly expressed in green tissue, which was confirmed by the fact that the protein was involved in chloroplast lipid biosynthesis. A similar pattern was observed for the soluble GPAT of sunflowers (HaPLSB) [23,56]. These authors demonstrated that the expression level of *HaPLSB* increased during cotyledon development, which was consistent with the increase in chloroplast membrane lipid biosynthesis rate in the early stage of plant growth, and remained at a high level in mature leaves. Group I has only one gene (AtS1) in most of the studied species, while in this study, there are six HvGPAT genes belonging to this category, of which five genes (HvGPAT5, HvGPAT9, HvGPAT15, HvGPAT18, and HvGPAT22) are grouped separately. The evolutionary history relationship (Figure 3B) also proves that these genes are the first batch of genes differentiated after the evolution of the hereditary GPAT.

It is reported that, compared with other members of the Arabidopsis GPAT family (GPAT1–8), Arabidopsis GPAT9 is more closely related to mammalian endoplasmic reticulum-localized GPAT3 and GPAT4, suggesting that the differentiation of the GPAT9 gene and GPAT1–8 of this species occurs before the evolutionary division between plants and mammals [50], and they have experienced different evolutionary patterns. Our phylogenetic analysis shows that GPAT9 is a very divergent branch. Studies on Arabidopsis have shown that reduced GPAT9 expression affects the number and composition of TAGs in seeds [51,52]. Evolutionary analysis showed that the genes differentiated from HvGPATs in group II were HvGPAT4, HvGPAT11, HvGPAT2, HvGPAT7, and HvGPAT21, respectively. Gene tissue expression analysis showed that the expression level of the barley GPAT gene in group Ⅱ was higher in embryo, root, cob, inflorescence, cotyledon, grain, and old leaf. In addition, all protein sequences classified as class I and GPAT9 have at least one putative transmembrane domain (TMD), indicating that they are membrane proteins.

The remaining GPATs (GPAT1–8) are comprised in group III. Studies have shown that members of Arabidopsis GPAT1–8 significantly affect the composition and quantity of cuticle or suberin [8]. Three different subclasses were observed in GPAT phylogeny within the third group (Figure 3A). The expansion and differentiation of these GPATs into different conserved subclasses (Figure 3A) are related to the key stages of the morphological and functional evolution of terrestrial plants. We observed that GPAT1 in subclass IIIa is closely related to GPAT2 and GPAT3. These GPATs have been proved to be located in mitochondria. Five of the GPATs found in barley belong to subclass IIIa (HvGPAT8, HvGPAT10, HvGPAT12, HvGPAT13, and HvGPAT17). The differentiation time is different, and the first differentiation gene is HvGPAT8. Tissue expression analysis showed that the expression levels of HvGPAT8 and HvGPAT17 genes in barley tissues was very low, the expression level of HvGPAT10 in roots was relatively high, and the expression levels of HvGPAT12 and HvGPAT13 in etiolated seedlings and palea were high, respectively. GPAT5 and GPAT7 also resulted from an independent and lineage-specific duplication event that appears to have occurred after monocot/eudicot divergence. GPAT5 is involved in suberin synthesis in roots and seeds [22]. Among the three genes in Ⅲb, HvGPAT1 differentiated first, followed by HvGPAT16 and HvGPAT20. The expression levels of these three genes in barley tissues was low. It is reported that GPAT4 and GPAT8 in IIIc are caused by an independent, lineage specific replication event in eudicotyledons. GPAT4 and GPAT8 are closely related to GPAT6. GPAT4 and GPAT8 are necessary for leaf cuticle formation [5,44], while GPAT6 is necessary for flower cuticle synthesis. In this study, the differentiation time of HvGPAT6, HvGPAT14, and HvGPAT19, belonging to class Ⅲc, was the latest. Among them, the expression level of *HvGPAT14* was higher in the developing grain, and the expression of the other two genes was lower. However, these GPATs exhibit variable gene expression patterns, indicating that they can be differentially regulated according to plant tissues.

### 3.2. Relationship between HvGPAT Gene and Plant Stress Resistance

Abiotic stresses, such as low temperature, drought, and salt stress, seriously affect the growth, development, and yield of plants. Studies have shown that gene expression patterns can provide useful clues for understanding gene function [14]. For example, *ZmGPAT4*, *ZmGPAT3*, and *ZmGPAT6* are highly expressed under low-temperature stress [12], while their rice lineal homologue *OsGPAT2* is highly expressed under low-temperature conditions [20]. In this study, *HvGPAT3*, *HvGPAT9*, and *HvGPAT15* are highly homologous to *AtS1*, and *HvGPAT8* and *HvGPAT17* are highly homologous to *AtGPAT1–3,* which all have low-temperature response elements. According to the results of the abiotic stress experiment, the expression level of *HvGPAT14* in leaves of group Ⅲc was significantly up-regulated under low-temperature stress; the expression levels of *HvGPAT5*, *HvGPAT8*, and *HvGPAT17* in leaves increased significantly under low-temperature stress; and ABA response element (ABRE) in these gene promoters mediated the response of plants to environmental stresses, such as high salt, drought, and low temperature [57]. It has been reported that MBS is the target of an MYB transcription factor to regulate plant frost resistance and drought resistance [58], suggesting that these *HvGPATs* containing MBS may be involved in the abiotic stress signal transmission of MYB. The content of ABA response elements in *HvGPATs* accounts for 90.91%, indicating that the *HvGPAT* genes may alleviate abiotic stress or participate in plant photomorphogenesis through the ABA pathway. The ABRE element is the most important hormone regulatory element in *HvGPATs*, which may mean that barley may rely on the ABA (abscisic acid) hormone pathway to adapt to more severe abiotic stress.

Barley exhibits resistance to various adverse conditions, such as pests and diseases, drought, and high salinity. Under salt stress, the expression of *HvGPAT14* is highly homologous to Arabidopsis *AtGPAT6,* and *HvGPAT4* is highly homologous to *AtGPAT9*. In addition, under salt stress, maize leaves and roots adopt different mechanisms to deal with salt stress [59]. During drought stress, leaf and root tissues share some common gene expressions, but the two tissues show different behavior in dehydration response [60]. In this study, the *HvGPAT* genes were expressed differently in roots and leaves. For example, under drought, cold, and salt stress, the expression level of *HvGPAT14* is different in roots and leaves, and the expression level in leaves is significantly up-regulated. This difference indicates that these genes may perform different functions in different tissues under salt stress or cold stress. The above results show that the relationship between the *HvGPAT* gene family and abiotic stress is relatively complex, which is consistent with the statement that the response mechanism of plants to abiotic stress is very complex [47]. At present, many studies have shown that overexpression of the GPAT gene in plants can improve the salt tolerance or cold tolerance of plants [33,36,37]. In our study, various physiological traits under abiotic stress were evaluated using transgenic Arabidopsis to gain further insight into the role of *HvGPAT18*, and it was found that transgenic seedlings have stronger resistance under abiotic stress than the WT plants. Therefore, the identification and specific expression gene screening of the *HvGPAT* gene family will not only help to deepen our understanding of the molecular mechanism of barley stress resistance but will also lay a foundation for the breeding and development of stress-resistant crops in the future.

## 4. Materials and Methods

### 4.1. Identification and Chromosomal Localization of GPAT Gene Family Members in Barley Genome

The completed genome sequences of barley were downloaded from the Ensembl database (http://www.plants.ensembl.org/index.html, accessed on 12 September 2021). The published GPAT protein sequences of Arabidopsis and rice were retrieved from the Arabidopsis Information Resource (TAIR release 10, http://www.arabidopsis.org, accessed on 12 September 2021) and the RGAP database (Rice Genome Annotation Project, http://rice.plantbiology.msu.edu/, accessed on 13 September 2021), respectively. Based on the GPAT protein sequence of Arabidopsis, BLASTP and HMM programs (E value = 0.0001) were used to search for GPAT candidate genes in the barley genome using Arabidopsis and rice GPAT protein sequences. Subsequently, the Pfam (http://pfam.sanger.ac.uk/, accessed on 12 September 2021), NCBI, CDD (http://www.omicsclass.com/article/310, accessed on 15 September 2021), and SMART (http://smart.emblheidelberg.de/, accessed on 15 September 2021) databases were applied to confirm each candidate member of the GPAT family with the PlsC-acyltransferase domain of Pfam (PF01553). Furthermore, the theoretical molecular weight (MW) isoelectric point (pI) and transmembrane domain region (TMD) of the GPAT proteins were predicted using the online ExPASy tool (http://web.expasy.org/, accessed on 15 September 2021), and subcellular localization was predicted using the online CELLO v2.5 server (http://cello.life.nctu.edu.tw/, accessed on 15 September 2021).

### 4.2. Multiple Sequence Alignment, Phylogenetic, and Evolutionary Analysis of Barley GPAT Protein

The prepared Arabidopsis and barley GPAT protein sequences were submitted to MEGA-X for multi-sequence alignment, and the output result file was opened with GeneDoc software (GeneDoc MFC Application, 2.7.0.0, accessed on 16 September 2021) to identify the conserved region.

To understand their evolutionary relationships, GPAT protein sequences from different plant species, including *Oryza sativa*, *Arabidopsis thaliana,* and *Hordeum vulgare*, were aligned with the Clustal W program. Subsequently, we constructed the maximum likelihood phylogenetic tree with 1000 bootstrap replicates using the poisson substitution model at default parameters in MEGA 7.0 software, and the resulting graph was visualized using the Evolview online service (http://www.omicsclass.com/article/963, accessed on 16 September 2021).

The prepared GPAT protein sequence was multisequence-aligned in MEGA-X software (10.2.6, accessed on 17 September 2021), and the sequence file and tree file in Newick format were saved. The evolution time of the species was calculated on the Time tree website (http://www.timetree.org/, accessed on 18 September 2021), the outgroup and calibration point were determined, and then the Reltime method of MEGA-Xwas used to infer the evolution time.

### 4.3. Chromosomal Localization, Gene Structure, and Protein Conserved Motif Analysis

All barley GPAT genes were mapped on the corresponding chromosomes according to their positional information provided in the barley genome annotation document. The chromosome location images of barley GPAT genes were portrayed by Mapchart software (v2.2, accessed on 20 September 2021) and AI tool (AI 2020cc, accessed on 20 September 2021).

Gene structure Display Server (http://GSDS.cbi.pku.edu.cn/, accessed on 20 September 2021) was used to display the genetic structure of *HvGPAT* genes. The MEME program (multiple for motif extraction, http://memesuite.org/tools/meme, accessed on 18 September 2021) was used to identify the conservative motif, and TBtools software (v2.029, accessed on 18 September 2021)was used for visual optimization.

### 4.4. Protein Three-Dimensional Structure Construction

The conserved sequences of Arabidopsis and barley GPAT proteins were submitted to SWISS-model (https://swissmodel.expasy.org/, accessed on 18 September 2021) online website to predict the three-dimensional structure of each protein and then the VMD software (1.9.4, accessed on 18 November 2022) was used for visualization.

### 4.5. Cis-Acting Elements Analysis

The 2 kb upstream DNA sequence of *HvGPAT* genes was intercepted and the cis-acting elements were analyzed using plantcare software (http://bioinformatics.psb.ugent.be/webtools/plantcare/html/ accessed on 18 September 2021).

### 4.6. Tissue Expression Profile Analysis

Barley RNA-Seq data were retrieved from the IPK database (https://www.ipk-gatersleben.de/en/, accessed on 24 September 2021). Heat maps of GPAT gene expression were generated based on their FPKM (Fragments per Kilobase Million) values using TBtools software (v2.029, accessed on 24 September 2021).

### 4.7. Plant Materials and Abiotic Stress Treatments for qPCR

The uniform and healthy seeds of barley ‘Morex’ (about 20 grains in commonly used Petri dishes) were selected, then surface sterilized for 10 min (no more than 10 min) with 30% hydrogen peroxide and rinsed 5–7 times with sterile water. The treated seeds were soaked in sterile water for 12 h, then transferred to the paper bed (the dish with wet filter paper) and arranged in an orderly manner with appropriate spacing. After germination in a dark environment for 48 h, they were transferred to the water culture system (modified Hoagland’s nutrient solution). The culture temperature was 20–22 °C (room temperature) with a 16 h photoperiod and 8 h dark period.

The barley variety ‘Morex’ was subjected to drought stress by application of 18% PEG-6000 to the nutrient solution, cold stress by treatment at 4 °C in the cryogenic incubator and salt stress by application of 200 mM NaCl to the nutrient solution for 48 h each. Root and leaf tissues samples were taken three times before application of the abiotic stress (14 days after germination), 2 days after the stress, and 2 days after recovery. The quantitative PCR primers are listed in Appendix A. Three biological replicates were performed for each sample and the relative expression levels of GPAT genes were calculated using the 2^−ΔΔCt^ method.

### 4.8. Arabidopsis Transformation, Plant Growth, and Abiotic Stress Treatments

The full-length coding sequence of *HvGPAT18* was recombined into the pCAMBIA1300-flag vector. The recombinant 35S:*HvGPAT18*-flag vector was used to transform *Agrobacterium tumefaciens* GV3101. *Agrobacterium tumefaciens* GV3101 harboring *35*S::*HvGPAT18* constructs was used to transform *A. thaliana* using the floral dip method [61]. Homozygous T3 transgenic Arabidopsis lines and Colombia (WT) plants were used for phenotypic and physiological analysis under abiotic stress.

To test the effects of different abiotic stresses on the seed germination rates and root lengths, approximately 30 seeds from each of the three selected T3 homozygous lines, as well as WT plants, were vernalized for 2 days at 4 °C, surface-sterilized, and subjected to drought, salt, and cold stress treatments with 200 mM mannitol, 130 mM NaCl, and 4 °C, respectively, and MS Medium under normal conditions were used as controls. The germination rate was calculated based on the percentage of seedlings that had reached the cotyledon stage at 10 days after sowing. Root lengths were measured when the seedlings were 20 days old.

Seven-day-old transgenic and WT seedlings grown on MS medium plates were transferred to pots filled with humus and watered well for 3 weeks; during the subsequent week, plants were irrigated at 2-day intervals with 200 mM NaCl or not watered, corresponding to salt and drought treatments, respectively. Plants that were treated with 4 °C corresponding to cold treatments. Plants that were well watered were used as a control. PRO content, and CAT (hydrogen peroxidase) activity was measured using a test kit (Jian cheng, Nanjing, China). All experiments were repeated three times.

## 5. Conclusions

In summary, 22 *HvGPAT* genes were identified in barley by a whole genome-wide analysis, and their characteristics and evolutionary relationships were elucidated through phylogenetic, conserved motif, and exon/intron structure analyses. Further analysis of the expression pattern of *HvGPATs* under cold, drought, and salt stress treatments revealed that *HvGPATs* responded to abiotic stress. Based on the performance of over-expressing Arabidopsis plants using the transgenic method, it was also confirmed that this *HvGPAT18* gene had potential tolerance functions to abiotic stress.

## Figures and Tables

**Figure 1 ijms-25-06101-f001:**
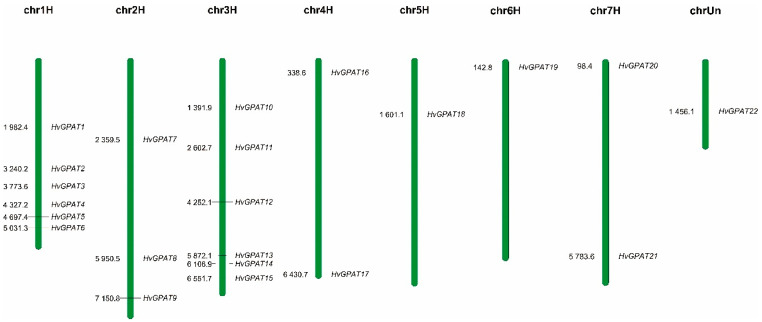
Chromosomal localization of the *HvGPAT* genes on the 7 barley chromosomes. Chromosome numbers were shown at the top of each vertical bar. On the left side of the chromosome is the physical location, and on the right side is the name of the HvGPAT genes (chr: chromosomes).

**Figure 2 ijms-25-06101-f002:**
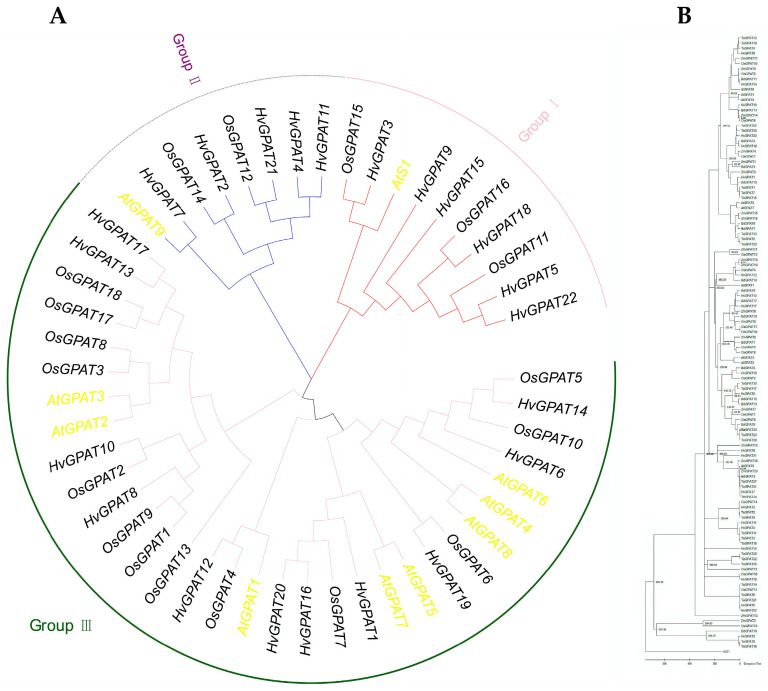
Phylogenetic and evolutionary analysis of HvGPAT proteins. (**A**) The phylogenetic tree was constructed in three taxa: barley (*Hordeum vulgare*), Arabidopsis (*Arabidopsis thaliana*), and rice (*Oryza sativa*) using the N-J method with 1000 bootstrap replicates. (**B**) Evolutionary map of GPAT genes in *Oryza sativa* (OsGPATs), *Brassica napus* (BdGPATs), *Zea may* (ZmGPATs), *Arabidopsis thaliana* (AtGPATs), and *Hordeum vulgare* (HvGPATs).

**Figure 3 ijms-25-06101-f003:**
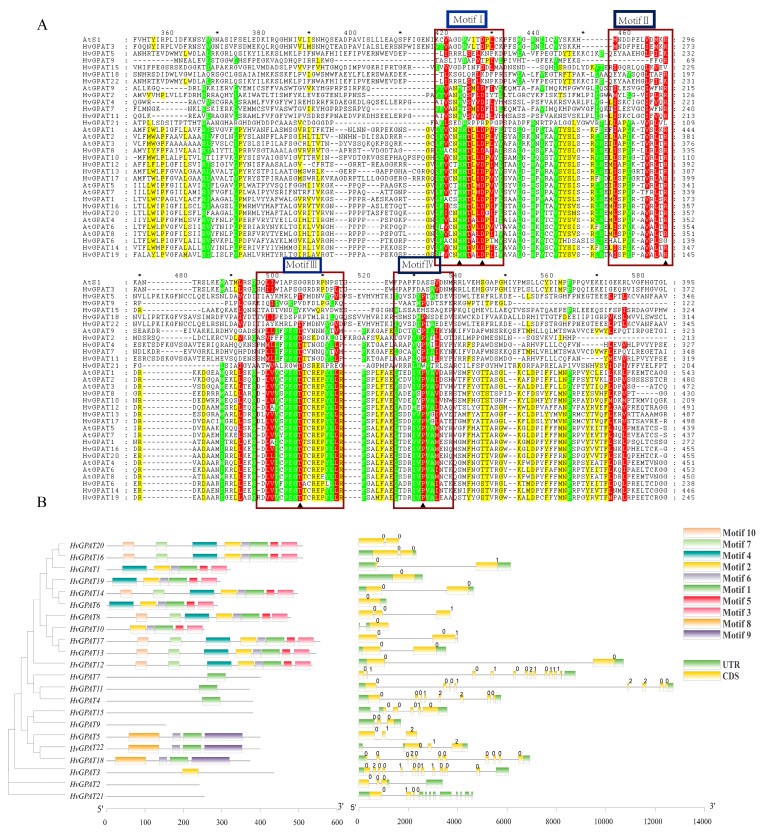
The protein sequence alignment, conserved motif, and gene structure of the HvGPAT genes. (**A**) Alignment of HvGPAT proteins and identification of conserved amino acid motifs. The multiple sequence alignment of the 22 GPAT proteins was performed in the CLUSTALX program. The four conserved acyltransferase amino acid motifs (Blocks I–IV) are boxed, and residues critical for enzyme catalysis and substrate binding are highlighted by a black triangle. Red, green and yellow in the sequence indicate that the sequence alignment is from high to low. * indicates consistent sequence. (**B**) Conserved motif and exon–intron gene structure of HvGPAT genes. The conserved motifs were identified by MEME. Motifs were indicated by different colored boxes with the motif number, while non-conserved sequences were represented by grey lines. Length of motifs was exhibited proportionally. Gene structure analysis was performed using the exon–intron length. The length of exons and introns of each HvGPAT gene was displayed proportionally. 0 1 2: intron phase.

**Figure 4 ijms-25-06101-f004:**
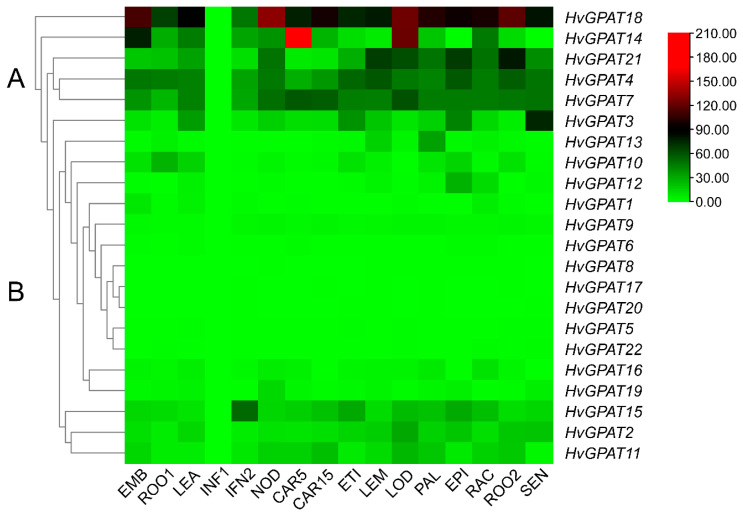
Expression profiles of the *HvGPAT* genes in different tissues. The RNA-seq data of 16 tissues in different developmental stages of barley seedlings were obtained from the BARLEX database at the Leibniz Institute of Plant Genetics and Crop Plant Research (IPK). X-axis: mRNA levels in 16 different tissues and life stages of barley; EMB: 4-day embryos; ROO1: roots from seedlings (10 cm shoot stage); LEA: shoots from seedlings (10 cm shoot stage); INF1: young developing inflorescences (5 mm); INF2: developing inflorescences (1–1.5 cm); NOD: developing tillers, 3rd internode (42 DAP); CAR5: developing grain (5 DAP); CAR15: developing grain (15 DAP); ETI: etiolated seedling, dark cond; LEM: inflorescences, lemma (42 DAP); LOD: inflorescences, lodicule (42 DAP); PAL: dissected inflorescences, palea (42 DAP); EPI: epidermal strips (28 DAP); RAC: inflorescences, rachis (35 DAP); ROO2: roots (28 DAP); and SEN: senescing leaves (56 DAP) were used to analyze tissues expression pattern. The expression level was shown in color as a scale: red represents high expression level and green represents low expression level. A and B stands for two different branches.

**Figure 5 ijms-25-06101-f005:**
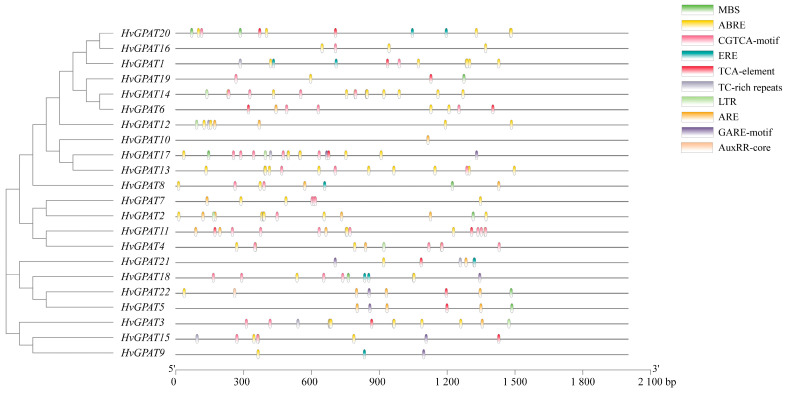
Prediction of cis-acting elements of the *HvGPAT* promoters. Many cis-acting elements were detected in the promoter region of each *HvGPAT* gene and different colors and shapes represent different cis-acting elements.

**Figure 6 ijms-25-06101-f006:**
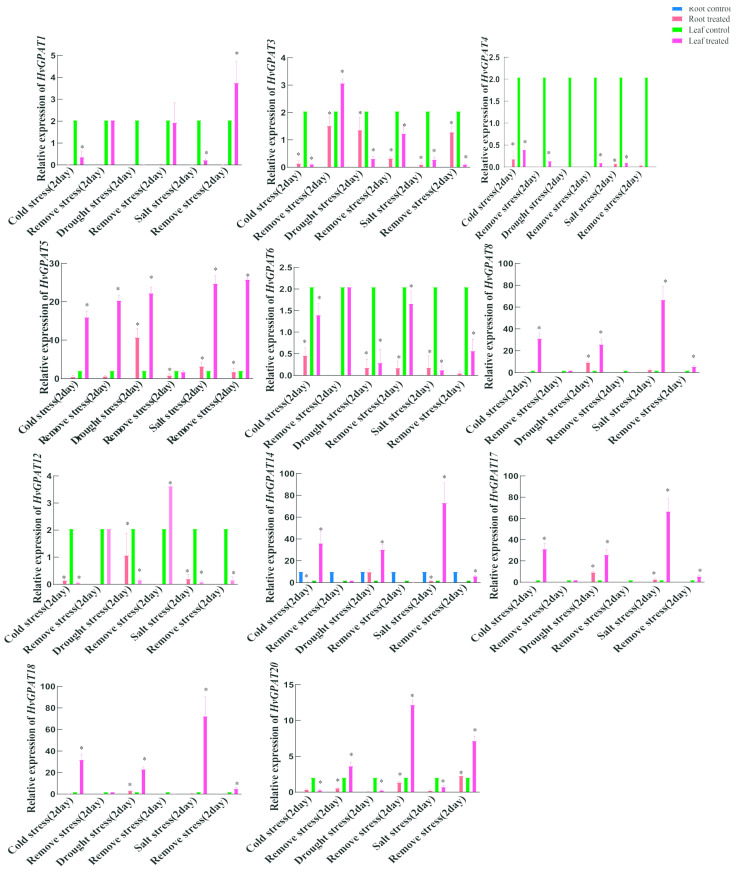
Expression profiles of 11 *HvGPAT* genes under different abiotic stresses. Data were normalized to the actin gene, and vertical bars indicate the standard deviation. The relative expression levels of the *HvGPAT* genes under different abiotic stresses were measured by qRT–PCR. The mean (±SE) expression values were calculated from three independent biological replicates and three technical replicates (* *p* < 0.01).

**Figure 7 ijms-25-06101-f007:**
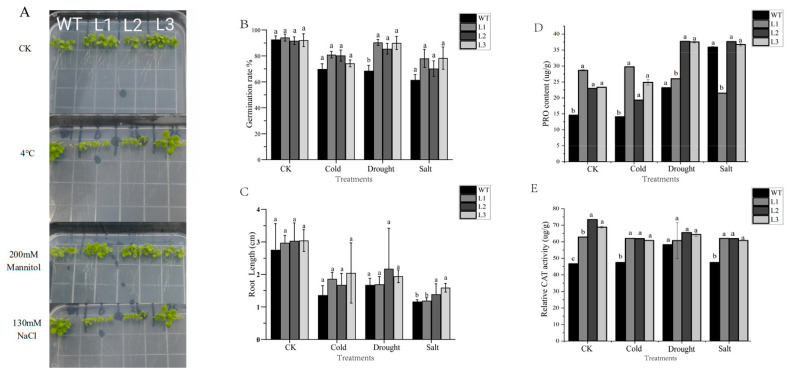
Effect of abiotic stress on phenotype and physiology of WT and *HvGPAT18* transgenic *Arabidopsis thaliana* lines. (**A**) Seedlings at 20 days after transfer to MS, MS with 4 °C, MS + 200 mM mannitol plates, or MS + 130mM NaCl, (**B**) germination rates, (**C**) Root length, (**D**) PRO (Proline) content, (**E**) CAT (hydrogen peroxidase) activity. Different lowercase letters indicate a significant difference at the 0.05 level; the same lowercase letters indicate a significant difference at the 0.05 level.

**Table 1 ijms-25-06101-t001:** Overview of the physicochemical characteristics and subcellular localization prediction of GPAT gene family in Barley.

Accession Number	Gene	Chro	Length	Molecular	Isoelectric	Int Loc ^b^	Transmembrane Region		
	Symbol	Dist ^a^	(Amino Acid)	Weight (Da)	Point			Instability Index	Grand Average of Hydropathicity (GRAVY)
HORVU1Hr1G050900.1	*HvGPAT1*	1H	436	48,648.90	7.31	ER	71-93-bp	44.14	0.178
HORVU1Hr1G059350.1	*HvGPAT2*	1H	381	41,946.90	9.23	Chl	13-35bp,106-128bp	42.61	0.295
HORVU1Hr1G044570.1	*HvGPAT3*	1H	242	26,958.60	10.2	Chl	no	50.46	−0.337
HORVU1Hr1G066030.1	*HvGPAT4*	1H	401	45,975.50	8.53	Chl, ER	no	47.11	−0.086
HORVU1Hr1G073340.1	*HvGPAT5*	1H	290	31,761.60	9.02	ER	53-75,90-107,346-365,369-391	45.78	0.040
HORVU1Hr1G032150.1	*HvGPAT6*	1H	323	35,143.70	9.82	ER	37-59	41.33	−0.007
HORVU2Hr1G081970.1	*HvGPAT7*	2H	480	52,043.10	10.09	ER, Chl	105-127,132-154	53.45	−0.213
HORVU2Hr1G045130.1	*HvGPAT8*	2H	401	46,548.80	9.97	ER, Mit	91-113,235-257	36.82	0.210
HORVU2Hr1G108810.1	*HvGPAT9*	2H	154	17,324	9.19	Mit	no	52.65	−0.312
HORVU3Hr1G080190.1	*HvGPAT10*	3H	545	59,545.20	9.59	Mit	no	42.25	−0.115
HORVU3Hr1G056830.1	*HvGPAT11*	3H	536	59,231.50	9.3	ER, Chl	74-96,131-148	52.42	−0.067
HORVU3Hr1G097070.2	*HvGPAT12*	3H	381	42,426.90	6.61	ER	5-24,285-307,312-334	40.49	0.169
HORVU3Hr1G029770.10	*HvGPAT13*	3H	255	28,932.30	9.27	ER	96-118	41.06	0.099
HORVU3Hr1G084990.1	*HvGPAT14*	3H	497	55,373.30	9.47	Mit	71-93,247-269	44.71	0.082
HORVU3Hr1G041560.1	*HvGPAT15*	3H	372	40,775.40	7.2	Chl, ER	no	49.47	−0.433
HORVU4Hr1G011110.1	*HvGPAT16*	4H	511	55,592.50	9.42	Mit	62-84,255-277	36.62	0.171
HORVU4Hr1G089610.1	*HvGPAT17*	4H	556	61,130.50	9.43	ER, Mit	no	38.34	0.041
HORVU5Hr1G027810.1	*HvGPAT18*	5H	374	42,490.80	10.3	ER	15-37,311-330,340-362	43.37	0.286
HORVU6Hr1G006810.1	*HvGPAT19*	6H	296	31,952.80	9.94	ER, Mit	45-67	40.47	0.092
HORVU7Hr1G095010.1	*HvGPAT20*	7H	255	28,011.70	10.07	Mit	61-83,254-276	34.45	0.218
HORVU7Hr1G007570.1	*HvGPAT21*	7H	511	56,464.90	9.04	ER, Mit	108-130	43.39	−0.162
HORVU0Hr1G027290.1	*HvGPAT22*	Un	400	45,872.30	8.6	ER	52-74,89-106,345-364,368-390	45.06	0.034

^a^ Chro dist: chromosomal distribution, Un: chromosomal location not determined. ^b^ Int loc: intracellular location, Chl: chloroplast, CM: cell membrane, ER: endoplasmic reticulum, Cyt: cytoplasmic.

## Data Availability

Data are contained within the article and Appendix A.

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
