# Peer review of "Genome-Wide Identification, Characterization, Evolutionary Analysis, and Expression Pattern of the GPAT Gene Family in Barley and Functional Analysis of HvGPAT18 under Abiotic Stress"

_ijms, 2024, doi:10.3390/ijms25116101_

Round 1

Reviewer 1 Report

Comments and Suggestions for Authors

This work entitled Genome-wide identification, characterization, evolution analysis and expression pattern of the GPAT gene family in barley and functional analysis of HvGPAT18 under abiotic stress by Chenglan et al. identified 22 glycerol-3-phosphoacyltransferase (GPAT) genes in barley genome using in silico methods. Authors grouped the genes using phylogenetic methods, structural properties (functional elements, motifs, isoelectric point, instability etc., ) and divergence. Although this work has interesting findings I have some concerns not just about its writing style, spacing, misspellings but important points in the methods section need to be addressed. For instance, DAP has not been explained, title and subtitles use lower or upper cases in mixed.  Please check the attached file for marks I made on the manuscript. Figures need to be checked and corrections need to be made. What is street in Figure 6?

However, please rewrite the materials sections using my suggestion below:

You did not provide any data or reference regarding the development of transgenic Arabidopsis. The floral dip method needs to be referenced and shortly explained  Without any data it is not possible to conclude the findings.

Tissue expression profile analysis was based on their FPKM values but what were the values?

You stated that 20 grains of barley cv Morex was selected, but what was your selection criteria?

What type of Polyethylen glycol was used?

Comments on the Quality of English Language

Please check attached marked manuscript. 

Author Response

Dear Editors:

Thanks for your letter and the comments on our manuscript entitled “Genome-wide Identification, characterization, evolution analysis and expression pattern of the GPAT gene family in barley and Functional Analysis of HvGPAT18 under abiotic stress” (ID: ijms-2956033). Those comments are all valuable and very helpful for revising and improving our manuscript, as well as important to guide our research. We have studied comments carefully and made corrections which we hope will meet the requirements.

Yours sincerely,

Rui-jun Duan

Comments and Suggestions for Authors

This work entitled Genome-wide identification, characterization, evolution analysis and expression pattern of the GPAT gene family in barley and functional analysis of HvGPAT18 under abiotic stress by Chenglan et al. identified 22 glycerol-3-phosphoacyltransferase (GPAT) genes in barley genome using in silico methods. Authors grouped the genes using phylogenetic methods, structural properties (functional elements, motifs, isoelectric point, instability etc., ) and divergence. Although this work has interesting findings I have some concerns not just about its writing style, spacing, misspellings but important points in the methods section need to be addressed.

Dear Reviewers:

I am very grateful to your comments for the manuscript. We tried our best to improve the manuscript and made some changes in it. These changes will not influence the content and framework of this manuscript. Based on your comment and requirement, we have made extensive modification from misspellings errors, stylistic errors, to the inconsistency in the original manuscript. In addition, we also revised the methods section. A revised manuscript with the correction sections marked in red was attached and for easy check/editing purpose.

We appreciate for Reviewers’ enthusiastic work, and hope that the correction will meet the requirements.

Once again, thank you very much for your comments and suggestions.

Yours sincerely,

Rui-jun Duan

Responds to the reviewer’s comments:

For instance, DAP has not been explained, title and subtitles use lower or upper cases in mixed. Please check the attached file for marks I made on the manuscript.

Figures need to be checked and corrections need to be made. What is street in Figure 6?

Response: We are very grateful to reviewer for reviewing the paper so carefully. Our answers to each of the 3 questions are as follows.

DAP (days after pollination) has been explained in line 254.

The lowercase or uppercase of the title and subtitles have been checked and separated.

The Figures has been checked and corrected. The ‘street’ in Figure 6 is a spelling error, corrected to ‘stress’.

Responds to the reviewer’s comments:

However, please rewrite the materials sections using my suggestion below:

You did not provide any data or reference regarding the development of transgenic Arabidopsis. The floral dip method needs to be referenced and shortly explained Without any data it is not possible to conclude the findings.

Tissue expression profile analysis was based on their FPKM values but what were the values?

You stated that 20 grains of barley cv Morex was selected, but what was your selection criteria?

What type of Polyethylen glycol was used?

Response: We are very grateful to reviewer for reviewing the paper so carefully. Our answers to each of the 3 questions are as follows.

1) Considering that the current transgenic method for Arabidopsis is a widely adopted method and the experimental method has been described, no further explanation has been provided. The floral dip method has also cited references [52]. Also, I am not quite sure what data needs to be provided. Figure 6 already shows the growth and physiological data of transgenic Arabidopsis plants. If other data is needed, we can provide it at any time.

2) FPKM means Fragments per Kilobase Million (in line 507), and the expression units of transcriptome data provided by the IPK website are expressed in FPKM values.

3) The selection principle for the 20 grain barley variety Morex is uniform and healthy, just to ensure the normal germination of the seeds. We added selection criteria in the line 509.

4)PEG-6000, and we have replaced it in the line 518.

Reviewer 2 Report

Comments and Suggestions for Authors

This article sheds light on the crucial role of GPAT in barley, particularly in the biosynthesis of TAG and its response to abiotic stress. The comprehensive analysis of 22 GPAT genes in the barley genome, including phylogenetic classification, conserved motif analysis, and structural elucidation, provides a solid foundation for understanding their evolutionary history and functional diversity. The findings from tissue expression profiling and qRT-PCR validation highlight the involvement of HvGPAT genes in abiotic stress responses, underscoring their importance in plant adaptation. Particularly noteworthy is the significant upregulation of HvGPAT18 under abiotic stress conditions, coupled with its subcellular localization in the endoplasmic reticulum, suggesting a specific role in stress tolerance mechanisms. The use of transgenic Arabidopsis to evaluate physiological traits further corroborates the functional significance of HvGPAT18 in enhancing stress resistance.

The objective needs to be more clarified, discussion must be linked with studied attributes to make a story. Take-home message is missing in the article, which should be added in the abstract and conclusion.

Overall, this study offers valuable insights into the evolutionary dynamics and functional diversity of the barley GPAT gene family, paving the way for future research aimed at deciphering the molecular mechanisms underlying plant stress responses.

Author Response

Dear Editors:

Thanks for your letter and the comments on our manuscript entitled “Genome-wide Identification, characterization, evolution analysis and expression pattern of the GPAT gene family in barley and Functional Analysis of HvGPAT18 under abiotic stress” (ID: ijms-2956033). Those comments are all valuable and very helpful for revising and improving our manuscript, as well as important to guide our research. We have studied comments carefully and made corrections which we hope will meet the requirements.

Yours sincerely,

Rui-jun Duan

Comments and Suggestions for Authors

This article sheds light on the crucial role of GPAT in barley, particularly in the biosynthesis of TAG and its response to abiotic stress. The comprehensive analysis of 22 GPAT genes in the barley genome, including phylogenetic classification, conserved motif analysis, and structural elucidation, provides a solid foundation for understanding their evolutionary history and functional diversity. The findings from tissue expression profiling and qRT-PCR validation highlight the involvement of HvGPAT genes in abiotic stress responses, underscoring their importance in plant adaptation. Particularly noteworthy is the significant upregulation of HvGPAT18 under abiotic stress conditions, coupled with its subcellular localization in the endoplasmic reticulum, suggesting a specific role in stress tolerance mechanisms. The use of transgenic Arabidopsis to evaluate physiological traits further corroborates the functional significance of HvGPAT18 in enhancing stress resistance.

The objective needs to be more clarified, discussion must be linked with studied attributes to make a story. Take-home message is missing in the article, which should be added in the abstract and conclusion.

Overall, this study offers valuable insights into the evolutionary dynamics and functional diversity of the barley GPAT gene family, paving the way for future research aimed at deciphering the molecular mechanisms underlying plant stress responses.

Dear Reviewers:

I am very grateful to your comments for the manuscript. We tried our best to improve the manuscript and made some changes in it. These changes will not influence the content and framework of this manuscript. Based on your comment and requirement, we have made modification to the abstract and discussion in the original manuscript. In addition, we also revised the objectives of this manuscript. A revised manuscript with the correction sections marked in red was attached as the supplemental material and for easy check/editing purpose.

We appreciate for Reviewers’ enthusiastic work, and hope that the correction will meet the requirements.

Once again, thank you very much for your comments and suggestions.

Yours sincerely,

Rui-jun Duan

Reviewer 3 Report

Comments and Suggestions for Authors

Thank you very much for sending me the manuscript titled “Genome-wide Identification, characterization, evolution analysis and expression pattern of the GPAT gene family in barley and Functional Analysis of HvGPAT18 under abiotic stress” for review. This manuscript firstly reports the in-silico identification of 22 GPAT genes belonging to an evolutionarily conserved gene family in the barley genome on the basis of similarity in sequences and genes/proteins structures to those already reported in Arabidopsis and rice. Tissue specific expression profiling of genes and promoter region analysis was also carried out computationally. The manuscript secondly showed through quantitative PCR analysis that HvGPAT genes were involved in abiotic stress responses. Constitutive overexpression of HvGPAT18 gene in Arabidopsis thaliana using CaMV 35S promoter has been shown to confer more resistance to abiotic stresses, including drought, cold and salt stresses.

Here are some of my major concerns about publishing the manuscript in IJMS.

1.      Why the expression of only 11 genes was checked through qPCR why not all 22 genes. The authors have not given any reason for or inclusion or exclusion criteria of genes selection for expression analysis.

2.      For finding the rates of germination, were the seeds for the tested overexpression lines and WT obtained in the same batch and maintained under the same conditions?

3.      Was the expression of HvGPAT18 checked on RNA and Protein levels in the overexpression lines?

4.      Were knockdown/knockout mutant lines of HvGPAT18 added to the experiment in order to ascertain the function of this gene in resistance to abiotic stresses. If the HvGPAT18 overexpression lines are indecently more resistant to abiotic stresses, its knockdown/knockout mutant lines should be more sensitive to these stresses.

5.      Arabidopsis tissue-specific endogenous promoter should have been used for generating lines with overexpression in the desired tissue/s. Unintended constitutive expression of genes may lead to the undesired phenotypes.

6.      No mechanism of action of the HvGPATs have been explored/suggested, that how these genes mediate stress resilience in barley.

7.      There are many grammatical and typo mistakes, the MS should be thoroughly revised for English language for better understanding of the MS.

8.      Line 4, 41, 49, 60 and 65, many references have been clustered together, individual references should be cited at the exact site inside or at the end of sentences.

9.      Gene names/symbols should be italicized uniformly throughout the MS

Here are some of my minor concerns about publishing the manuscript in IJMS.

1.      Change all “Streets” to “stress” in figure 6

2.      Line 211, remove acyltrasferase

3.      Line 212, change figure 4 to figure 3

4.      Line 220, change figure S1 to S2

5.      Lines 285 to 290, need to rewrite the results of the experiment.

6.      Line 306. change figure S2 to S1

7.      Figure S2 and S3 repetition of the same figure, Figure 3 not cited in MS

8.      Line 321, correct typo phenotypic and physiological

9.      Line 433, correct grammatic/typo “the HvGPATs gene was expressed differently in roots

10.  Line 444, correct “than to the WT”

Regards 

Author Response

Dear Editors:

Thanks for your letter and the comments on our manuscript entitled “Genome-wide Identification, characterization, evolution analysis and expression pattern of the GPAT gene family in barley and Functional Analysis of HvGPAT18 under abiotic stress” (ID: ijms-2956033). Those comments are all valuable and very helpful for revising and improving our manuscript, as well as important to guide our research. We have studied comments carefully and made corrections which we hope will meet the requirements.

Yours sincerely,

Rui-jun Duan

Comments and Suggestions for Authors

Thank you very much for sending me the manuscript titled “Genome-wide Identification, characterization, evolution analysis and expression pattern of the GPAT gene family in barley and Functional Analysis of HvGPAT18 under abiotic stress” for review. This manuscript firstly reports the in-silico identification of 22 GPAT genes belonging to an evolutionarily conserved gene family in the barley genome on the basis of similarity in sequences and genes/proteins structures to those already reported in Arabidopsis and rice. Tissue specific expression profiling of genes and promoter region analysis was also carried out computationally. The manuscript secondly showed through quantitative PCR analysis that HvGPAT genes were involved in abiotic stress responses. Constitutive overexpression of HvGPAT18 gene in Arabidopsis thaliana using CaMV 35S promoter has been shown to confer more resistance to abiotic stresses, including drought, cold and salt stresses.

Dear Reviewers:

I am very grateful to your comments for the manuscript. We tried our best to improve the manuscript and made some changes in it. These changes will not influence the content and framework of this MS. Based on your comment and requirement, we have made extensive modification from errors, omissions of words, stylistic errors, to the inconsistency in the MS. In addition, we also revised the objectives of this MS. A revised manuscript with the correction sections marked in red was attached as the supplemental material and for easy check/editing purpose.

We appreciate for Reviewers’ enthusiastic work, and hope that the correction will meet the requirements.

Once again, thank you very much for your comments and suggestions.

Yours sincerely,

Rui-jun Duan

Responds to the reviewer’s comments:

We are very grateful to reviewer for reviewing the paper so carefully. Our answers to each of the 19 questions are as follows.

Here are some of my major concerns about publishing the manuscript in IJMS.

  1. Why the expression of only 11 genes was checked through qPCR why not all 22 genes. The authors have not given any reason for or inclusion or exclusion criteria of genes selection for expression analysis.

Response: The selection of 11 genes takes into account two aspects. Firstly, classification should be considered to ensure selection in each subgroup. Secondly, transcriptome data provided by IPK should be considered to select genes with higher expression levels as much as possible. In addition, reference should be made to select genes from subgroups related to abiotic stress as much as possible.

  1. For finding the rates of germination, were the seeds for the tested overexpression lines and WT obtained in the same batch and maintained under the same conditions?

Response: Yes, we obtained Arabidopsis seeds from the same batch and conducted germination rate studies on overexpressed lines and WT seeds under the same conditions.

  1. Was the expression of HvGPAT18 checked on RNA and Protein levels in the overexpression lines?

Response: We used qRT-PCR to detect the high expression level of HvGPAT18 in overexpressing lines at the RNA level.

  1. Were knockdown/knockout mutant lines of HvGPAT18 added to the experiment in order to ascertain the function of this gene in resistance to abiotic stresses. If the HvGPAT18 overexpression lines are indecently more resistant to abiotic stresses, its knockdown/knockout mutant lines should be more sensitive to these stresses.

Response: The HvGPAT18 gene comes from barley, so knockdown/knockout mutant lines cannot be obtained in Arabidopsis. The barley knockout/knockout mutant line of HvGPAT18 is currently being studied.

  1. Arabidopsis tissue-specific endogenous promoter should have been used for generating lines with overexpression in the desired tissue/s. Unintended constitutive expression of genes may lead to the undesired phenotypes.

Response: Thank you for your suggestion. Transgenic research on HvGPAT promoters is also underway.

  1. No mechanism of action of the HvGPATs have been explored/suggested, that how these genes mediate stress resilience in barley.

Response: Yes, this is a good suggestion, but due to research progress, the mechanism of action of HvGPATs cannot be proposed yet, and further research is needed on how these genes mediate abiotic stress resistance in barley.

  1. There are many grammatical and typo mistakes, the MS should be thoroughly revised for English language for better understanding of the MS.

Response: We have made revisions to grammar and spelling errors, hoping to help better understand MS.

  1. Line 4, 41, 49, 60 and 65, many references have been clustered together, individual references should be cited at the exact site inside or at the end of sentences.

Response: Okay, it has been modified.

  1. Gene names/symbols should be italicized uniformly throughout the MS

Response: Okay, it has been modified.

Here are some of my minor concerns about publishing the manuscript in IJMS.

Response:

  1. Change all “Streets” to “stress” in figure 6

Response: Okay, it has been modified.

  1. Line 211, remove acyltrasferase

Response: Okay, it has been modified.

  1. Line 212, change figure 4 to figure 3

Response: Okay, it has been modified.

  1. Line 220, change figure S1 to S2

Response: Okay, it has been modified.

  1. Lines 285 to 290, need to rewrite the results of the experiment.

Response: Okay, it has been modified.

  1. Line 306. change figure S2 to S1

Response: Okay, it has been modified.

  1. Figure S2 and S3 repetition of the same figure, Figure 3 not cited in MS

Response: Okay, it has been modified.

  1. Line 321, correct typo phenotypic and physiological

Response: Okay, it has been modified.

  1. Line 433, correct grammatic/typo “the HvGPATs gene was expressed differently in roots”

Response: Okay, it has been modified.

  1. Line 444, correct “than to the WT”

Response: Okay, it has been modified.
